# Effects of Altitude and Continuous Cropping on Arbuscular Mycorrhizal Fungi Community in *Siraitia grosvenorii* Rhizosphere

**Limin Yu** [1,2], **Zhongfeng Zhang** [1,*], **Longwu Zhou** [1] **and Kechao Huang** [1]

[1] Guangxi Key Laboratory of Plant Conservation and Restoration Ecology in Karst Terrain, Guangxi Institute of Botany, Guangxi Zhuang Autonomous Region and Chinese Academy of Sciences, Guilin 541006, China; yulimin1998@163.com (L.Y.); longzhouwu@163.com (L.Z.); kchuanggxib@163.com (K.H.)

[2] College of Life Sciences, Guangxi Normal University, Guilin 541006, China

* Correspondence: zfzhang@gxib.cn

**Abstract:** *Siraitia grosvenorii*, a medicinal plant with continuous cropping, is cultivated in southern China. Changes in the soil microbial community during continuous cropping can cause soil-borne diseases in *S. grosvenorii*. This experimental study aimed to determine the differences in the arbuscular mycorrhizal fungi (AMF) community structure and root colonization in the rhizosphere soil of *S. grosvenorii* with different continuous cropping years and altitudes. We tested three altitude gradients (low, 200–300 m; middle, 500–600 m; and high, 700–800 m) and four continuous cropping years (1, 2, 3, and 5 years). AMF colonization, along with AMF spore density, and the soil physicochemical properties of *S. grosvenorii* roots at different altitudes and continuous cropping years were determined. Illumina high-throughput sequencing was used to determine the molecular diversity of AMF in the rhizosphere of *S. grosvenorii* as they exhibited a symbiotic relationship. The AMF species in the rhizosphere soil of *S. grosvenorii* included 28 species of nine genera, including *Glomus, Claroideoglomus, Acaulospora, Paraglomus, Ambispora,* and so on. With an increasing altitude, the AMF colonization of *S. grosvenorii* roots increased significantly ($p < 0.01$); available phosphorus (AP) content was negatively correlated with AMF colonization ($p < 0.01$). *Glomus* and *Paraglomus* were the common dominant genera in the rhizosphere soil of *S. grosvenorii* planted for 2–5 years at a low altitude and 1 year at middle and high altitudes. The average relative abundance of *Glomus* increased with increasing continuous cropping years and altitude in the low-altitude and 1-year *S. grosvenorii* areas, respectively. Slightly acidic rhizosphere soil contributed to AMF colonization and improved the richness and diversity of the AMF community. Our results showed that altitude, AP, and pH are essential factors for predicting AMF infection and community changes in the *S. grosvenorii* rhizosphere. Here, Illumina high-throughput sequencing was used to study the species resources and community composition of mycorrhizal fungi in *S. grosvenorii* in the hilly areas of Guangxi, China. This study provides a theoretical basis for the application and practice of mycorrhizal fungi including the isolation and screening of dominant strains, inoculation of mycorrhizal fungi, and exploration of the effects of mycorrhizal fungi on the growth and active ingredients of medicinal plants.

**Keywords:** altitude; continuous cropping; soil factors; *Siraitia grosvenorii*; arbuscular mycorrhizal fungi

## 1. Introduction

Symbiotic fungi are essential for promoting plant growth and development, enhancing stress resistance, and improving the content and quality of secondary metabolites in forests, agriculture, and ecological neighborhoods [1]; as such, they have become a popular research topic. For example, arbuscular mycorrhizal fungi (AMF) are soil fungi that form mutualistic symbionts with most plants [2]; their colonization ability in different plant types ranges from 80% to 90% [3]. Among other functions, AMF help plants obtain 50–75% and 30% of their phosphorus [4,5] and nitrogen requirements, respectively [5,6]. Host plants provide

30% of the carbon requirements for the growth of mycorrhizal fungi [7]. AMF and host plants form an ecological resource transfer and a two-way nutrient-exchange symbiotic relationship, thereby enabling plants to respond positively in low-fertility environments.

Understanding the dynamic and variable driving factors of AMF community changes is essential for the application of AMF biological functions in various fields. In regional habitats, the availability and mobility of soil pH, nitrogen, phosphorus, potassium, other elements, and restrictive nutrients directly or indirectly affect AMF community composition and diversity [8–10]. A significant altitude gradient leads to differences in geographical and ecological factors, such as temperature, humidity, precipitation, microclimate characteristics, soil properties [11–13], and biodiversity in mountainous areas [11]. These differences affect the establishment of soil AMF communities, species distribution, survival strategies, species similarity, and the number of propagules [11]. The long-term continuous cropping of plants changes the soil's physicochemical properties, disrupts the balance of the microbial community structure, transforms soil types from "bacterial" to "fungal," and changes soil microbial community composition and diversity [14]. Therefore, determining the correlation between altitude, continuous cropping years, soil nutrients, and mycorrhizal fungal community composition and diversity is crucial.

*Siraitia grosvenorii* (Swingle) C. Jeffrey ex Lu et Z. Y. Zhang is a medicinal and edible plant of Cucurbitaceae, which has the effect of moistening the lungs and relieving coughing [15]. The planting areas and yield of *S. grosvenorii* in northern Guangxi (Longsheng, Yongfu, and other areas) contribute to 95% of the total in China [16]. However, the continuous planting of *S. grosvenorii* for over 2 years leads to continuous cropping obstacles and threatens plant growth because continuous cropping affects plant rhizosphere soil and soil microorganisms. AMF are beneficial microorganisms for plant growth because they can improve the microenvironment of plant roots, absorption of nutrients, and resistance to stress, and they can alleviate the damage caused by continuous cropping [1,5,7]. Therefore, the decrease in AMF diversity and root colonization in the plant rhizosphere is a challenge for continuous cropping in various plants, including *S. grosvenorii*.

The incidence of the continuous cropping obstacle of *S. grosvenorii* is low in mountainous areas at an altitude of >700 m. In contrast, the continuous cropping obstacles of *S. grosvenorii* in mountainous areas at an altitude of <700 m are higher, which may be related to the influence of altitude on the diversity and richness of AMF in rhizosphere soil. Therefore, this study explored the mycorrhizal fungi resources of *S. grosvenorii* for the first time in the spatial structure of multi-functional plant farmlands, such as hilly areas, with small altitude gradients in Guangxi, China. In this study, the correlation between rhizosphere biodiversity and the environment of Siraitia grosvenorii was explored for the first time by combining non-biological factors such as continuous cropping years, altitude, and soil factors with biological factors such as AMF germplasm resources, community structure characteristics, and root colonization. This study provides a reference for the exploration of multi-functional medicinal plant mycorrhizal fungi germplasm resources, and provides basic data for the subsequent screening of dominant strains, mycorrhizal fungi inoculation, and the application of mycorrhizal fungi ecological functions to future plant cultivation practices.

## 2. Materials and Methods

### 2.1. Experimental Design

The area of a >700 m altitude above sea level in Longsheng County, Guangxi, China is an alpine mountainous region, as snowfall and freezing occur during winters. Longsheng County (Guangxi, China; 109°43′–110°21′ E, 25°29′–26°12′ N) has an average annual temperature of 18–19 °C, average annual rainfall of 1500–2400 mm, and average frost-free period of 314 days; it is located in a subtropical monsoon climate zone and receives abundant rainfall [15]. This study selected *S. grosvenorii* 'Dadi No.1.', which has a large planting area in Guilin (Guangxi, China), as the experimental variety.

To test the effects of altitude and cropping years, four study sites were selected, namely Shangtang Village (109°52′ E, 25°50′ N) of Piaoli Town, Jinjie Village (110°1′ E, 25°46′ N) of Longsheng County, and Baozeng Village (109°50′ E, 25°59′ N) and Diling Village (109°50′ E, 25°57′ N) of Lejiang County. The linear geographical distance of each planting base was 17.3 km from Shangtang Village to Jinjie Village, 16.8 km from Shangtang Village to Baozeng Village, 13.3 km from Shangtang Village to Diling Village, 30.3 km from Jinjie Village to Baozeng Village, 27.5 km from Jinjie Village to Diling Village, and 3.7 km from Diling Village to Baozeng Village. The soil at each planting base was a relatively uniform loam or sandy loam [17], ensuring similar site conditions for all plots. Three altitudinal gradients and four continuous cropping years were used (Table 1). The planting area for each treatment was approximately 350 m$^2$, and the planting density was 130 plants/acre.

**Table 1.** Soil physicochemical properties of sampling points.

| Sampling Site | Sample Code | Altitude (m) | Continuous Cropping Years (a) | Number of Samples | pH | SOM (g kg$^{-1}$) | AN (g kg$^{-1}$) | AP (mg kg$^{-1}$) | AK (mg kg$^{-1}$) |
|---|---|---|---|---|---|---|---|---|---|
| Shangtang | A1 | 248 | 1 | 5 | 4.52 ± 0.3 | 34 ± 7 | 0.41 ± 0.14 | 605 ± 134 | 657 ± 94 |
| Jinjie | A2 | 259 | 2 | 5 | 5.01 ± 0.5 | 36 ± 6 | 0.35 ± 0.04 | 596 ± 207 | 971 ± 436 |
| Shangtang | A5 | 248 | 5 | 5 | 5.29 ± 0.2 | 40 ± 3 | 0.30 ± 0.03 | 359 ± 267 | 280 ± 154 |
| | B1 | 513 | 1 | 5 | 5.42 ± 0.6 | 47 ± 12 | 0.28 ± 0.04 | 353 ± 249 | 325 ± 120 |
| Baozeng | B2 | 561 | 2 | 3 | 4.64 ± 0.2 | 48 ± 11 | 0.30 ± 0.03 | 233 ± 89 | 393 ± 138 |
| | B3 | 513 | 3 | 3 | 4.99 ± 0.5 | 60 ± 1 | 0.42 ± 0.02 | 373 ± 47 | 122 ± 42 |
| | C1 | 762 | 1 | 5 | 4.79 ± 0.3 | 87 ± 10 | 0.47 ± 0.06 | 193 ± 72 | 376 ± 91 |
| Diling | C2 | 763 | 2 | 5 | 5.20 ± 0.4 | 74 ± 9 | 0.91 ± 1.01 | 207 ± 106 | 1013 ± 723 |

Data are presented as the mean ± standard deviation. In the sample code column, A, B, and C represent low (200–300 m), middle (500–600 m), and high (700–800 m) altitudes, respectively; the numbers represent continuous cropping years. A1, A2, and A5 represent low-altitude (200–300 m) continuous cropping for 1, 2, and 5 years, respectively. B1, B2, and B3 represent continuous cropping for 1, 2, and 3 years at a medium altitude (500–600 m), respectively. C1 and C2 represent 1- and 2-year continuous cropping at a high altitude (700–800 m), respectively. "m" and "a" are the measurement units for altitude and continuous cropping years, respectively. SOM: soil organic matter. AN: available nitrogen. AP: available phosphorus. AK: available potassium.

Continuous cropping years were set as the first year of planting (plot had not been planted with *S. grosvenorii* or other plants before), second year of planting (plot had been planted with *S. grosvenorii* for 2 consecutive years, but had not been planted with *S. grosvenorii* or other plants previously), third year of planting (plot had been planted with *S. grosvenorii* for 3 consecutive years but had not been planted with *S. grosvenorii* or other plants previously), and fifth year of planting (plot had been planted with *S. grosvenorii* for 5 consecutive years but had not been planted with *S. grosvenorii* or other plants previously).

Low-altitude (group A, 200–300 m) plots were those located in the Shangtang and Jinjie villages. Shangtang Village was set at 1-year and 5-year continuous cropping treatments (represented by A1 and A5, respectively). Jinjie Village was set at 2-year continuous cropping treatments (represented by A2). Middle-altitude (group B, 500–600 m) plots were those located in Baozeng Village, and were set at 1-year, 2-year, and 3-year continuous cropping treatments (represented by B1, B2, and B3, respectively). High-altitude (group C, 700–800 m) plots were those located in the Baozeng and Diling villages, for which 1-year and 2-year continuous cropping treatments were set (C1 and C2, respectively).

Five replicates were set for A1, A2, A5, B1, C1, and C2, and three replicates were set for B2 and B3. Planting methods and management measures were the same for all plots; before sowing, each planting hole of *S. grosvenorii* was applied with a 1500 g sheep manure organic fertilizer. After 30 days of planting, 500 g of a phosphate fertilizer and 250 g of a potassium fertilizer were applied, and 250 g of a potassium fertilizer was applied after 45 days of planting.

### 2.2. Sampling and Processing

Root and rhizosphere soil samples of *S. grosvenorii* were collected in mid-November 2022. The plant rhizosphere refers to the region where plant roots exchange substances

with the external environment; that is, the contact surface of interaction among plants, soil, and microorganisms [18]. Rhizosphere soil refers to the soil attached to this interface [18]. During sampling, the litter layer and large sand on the soil surface were removed, and a clean iron shovel was used to dig down to the plant roots. The roots of the whole plant were unearthed as much as possible, and the aboveground parts were removed to retain fresh, fine, and tough roots [19]. A total of 1.5 kg of soil samples was collected from 0–30 cm of the soil layer. Three *S. grosvenorii* roots and their rhizosphere soil samples were randomly selected from each treatment and mixed to form biological composite samples. In total, 36 samples of *S. grosvenorii* roots and rhizosphere soil were collected. The fresh roots were washed with sterile water and placed in an FAA fixative ($CH_2O$, 130 mL + $CH_3COOH$, 50 mL + 50% $C_2H_5OH$, 2000 mL) for AMF infection structure observation and colonization determination. Rhizosphere soil samples were divided into two parts. One portion was preserved on dry ice and subsequently transferred for DNA extraction and molecular diversity detection. The other portion was homogenized after air drying at 25 °C for soil physicochemical properties and an AMF spore density analysis.

*2.3. Determination of Physicochemical Properties of Rhizosphere Soil*

The soil was sieved with soil sieves (2 and 0.15 mm) after drying at 25 °C. The soil pH was measured using a calibrated pH meter (FE28, Shanghai, China), and the soil-to-water ratio was 1:5 (*w/v*). The soil organic matter (SOM) content was determined using the $K_2Cr_2O_7$/$FeSO_4$ method. Under heating conditions, the organic carbon in soil was oxidized with a quantitative $K_2Cr_2O_7$–$H_2SO_4$ solution, and titrated with an $FeSO_4$ solution. According to the milliliter of $FeSO_4$ titration and the oxidation correction coefficient, the amount of organic carbon was calculated, and then multiplied by the conversion coefficient, which was the SOM content. Soil samples were added with an $FeSO_4$ reducing agent, and available nitrogen (AN) in soil was determined with an automatic Kjeldahl nitrogen analyzer (K1160, Shandong, China), including ammonium nitrogen, nitrate nitrogen, amino acids, amides, and the sum of nitrogen in easily hydrolyzed proteins [20]. Soil available phosphorus (AP) was extracted with HCl/$H_2SO_4$, centrifuged, and filtered; added to a chromogenic agent; and determined using an ultraviolet-visible spectrophotometer (UV-1800PC, Shanghai, China). Soil available potassium (AK) content was determined using an $NH_4OAc$ solution. $NH_4^+$ is exchanged with $K^+$ on the surface of the soil colloid. The replaced $K^+$ and water-soluble $K^+$ enter the solution together. The K in the leaching solution can be directly determined with a flame photometer (FP640, Shanghai, China; Table 1) [21].

*2.4. Determination of AMF Colonization and Spore Density*

The root segments in the FAA fixative were cut, made transparent with 10% (*w/v*) KOH, decolored with 10% $H_2O_2$, acidified with 2% HCl, and stained with 0.05% trypan blue [19]. The presence of AMF hyphae, vesicles, and other structures in the roots of *S. grosvenorii* was regarded as AMF colonization in the plant. Thirty root segments were selected for each sample, and the mycorrhizal structures were observed and photographed under a biological microscope (Leica, Wetzlar, Germany) at ×40 magnification. Colonization (%) = the number of root segments with the mycorrhizal structure/the number of observed root segments [19]. In addition, AMF spores were isolated from 25 g of dry rhizosphere soil using wet sieve precipitation [19]. The spores were observed under a ×10 stereomicroscope (Shanghai, China), and the number of spores were calculated.

*2.5. Determination of AMF Molecular Diversity in S. grosvenorii Rhizosphere*

Genomic DNA were extracted from soil samples using a MagaBio Soil Kit (BSC48S1E, BIOER, Hangzhou, China) according to the manufacturer's instructions, and the purity and concentration of the DNA were detected using NanoDrop One (Thermo Fisher Scientific, Waltham, MA, USA). AMV4.5NF (5'-AAGCTCGTAGTTGAATTTCG-3') and AMDGR (5'-CCCAACTATCCCTATTAATCAT-3') primers [22] and Takara Premix Taq® Version 2.0 (TaKaRa Biotechnology Co., Dalian, China) were used for polymerase chain reaction (PCR)

amplification. The PCR amplification system comprised 25 μL of 2 × Premix Taq, 1 μL of Primer-F (10 μM), 1 μL of Primer-R (10 μM), 50 ng of DNA, and nuclease-free water added to 50 μL. PCR reaction conditions were as follows: 94 °C for 5 min, 94 °C for 30 s, 52 °C for 30 s, 72 °C for 30 s, with a total of 30 cycles, and a subsequent reaction at 72 °C for 10 min, followed by preservation at 4 °C. The purity of PCR products was detected using 1% agarose gel electrophoresis. The PCR mixture was recovered using an EZNA ® Gel Extraction Kit (Omega, Norcross, GA, USA), and the target DNA fragment was recovered using TE buffer elution. The library was constructed according to the standard procedure of the NEBNext® UltraTM II DNA Library Prep Kit for Illumina® (New England Biolabs, Ipswich, MA, USA). The amplicon library was sequenced using the Illumina Nova 6000 platform. Sequencing was performed by Guangdong Magigene Biotechnology Co., Ltd. (Guangzhou, China).

## 3. Data Processing

### 3.1. Sequencing Data Processing

Fastp (v 0.14.1, https://github.com/OpenGene/fastp, accessed on 14 February 2023) was used to perform sliding-window quality clipping (parameters W4-M20) on the two-terminal raw read data. Based on the primer information at both ends of the sequence, the primers were removed using Cutadapt (v.1.14, https://github.com/marcelm/cutadapt/, accessed on 14 February 2023) to obtain clean paired-end reads after quality control. According to the overlapping relationship between PE reads, operational taxonomic units (OTU) splicing was performed using usearch-fastq_ merge pairs (v 10.0.240, http://www.drive5.com/usearch/, accessed on 14 February 2023). OTU clustering was performed using usearch (v 10.0.240). Sequences with a similarity of ≥97% were classified as the same OTU. OTU species annotation was performed using usearch-sintax (v 10.0.240) and compared with the UNITE (v 8.0, https://unite.ut.ee/, accessed on 14 February 2023) database to obtain species annotation information.

### 3.2. Statistical Analysis

After the PCR detection of 36 soil samples, the C2.2 sample had no target band, and it was determined as unqualified after quality inspection. This sample was, therefore, removed from subsequent analyses, resulting in 35 samples. Usearch-alpha_div (v 10.0.240, http://www.drive5.com/usearch/, accessed on 14 February 2023) was used to calculate richness; the Chao1, Shannon–Wiener, and Simpson indices were used to determine community richness and diversity. Excel and SPSS (v 26.0) were used to analyze AMF colonization, spore density, alpha diversity, correlation, and the significance of environmental factors. R (v 4.2.3) was used to draw a histogram of AMF colonization, spore density, and the relative abundance map of the AMF genus level in each sample. A non-metric multidimensional scaling (NMDS) analysis was performed using the vegan package in R (v 4.2.3) based on Bray–Curtis and Spearman coefficients. Analyses of molecular variance (AMOVA) and similarities (ANOSIM) were used to compare group differences. A Venn diagram was drawn using the R software to reveal the number of common and unique OTUs. The linear discriminant analysis effect size (LEfSe) was used to analyze the differences in species between groups. A linear discriminant analysis (LDA) score of ≥2 was considered a biomarker. A detrended correspondence analysis (DCA), a canonical correspondence analysis (CCA), and Mantel test analyses were performed in the R software to test the relationships between environmental factors and the microbial community structure. Heatmaps were drawn to reveal the correlations between environmental factors and AMF genera.

## 4. Results

### 4.1. Physicochemical Properties of Rhizosphere Soil of S. grosvenorii

The physicochemical properties of the soils at different altitudes and continuous cropping years are listed in Table 1. The soils of all samples were weakly acidic (pH 4.23–6.14); 97% of the soil samples had a pH of <6.00, with an average of 5.00 ± 0.5. The average

SOM content was 53.35 ± 20 g kg$^{-1}$, AN content was 0.436 ± 0.40 g kg$^{-1}$, AP content was 371.79 ± 223 mg kg$^{-1}$, and AK content was 545.91 ± 432 mg kg$^{-1}$. With an increase in altitude, the SOM content in the rhizosphere soil of *S. grosvenorii* increased; the AP content decreased after the first planting and continuous cropping for 2 years. With an increase in continuous cropping years in low-altitude areas, SOM accumulation increased, and AP content decreased.

### 4.2. AMF Colonization, and AMF Spore Density of S. grosvenorii

AMF were detected in the root samples collected from *S. grosvenorii*. Typical structures included AMF hyphae, vesicles indicating a significant mycorrhizal symbiotic relationship between *S. grosvenorii* and AMF. The colonization of AMF in the roots and spore density in the rhizosphere soil of *S. grosvenorii* are shown in Figure 1. The average colonization of AMF was 57.33% and the AMF average spore density was 3.37/g. The average colonization of AMF in the roots of *S. grosvenorii* showed an increasing trend with an increasing altitude in the following order: low altitude (45.33%) < middle altitude (62.12%) < high altitude (72.33%). The average spore density of AMF decreased with an increasing altitude in the following order: low altitude (4.09/g) > middle altitude (2.95/g) > high altitude (2.89/g).

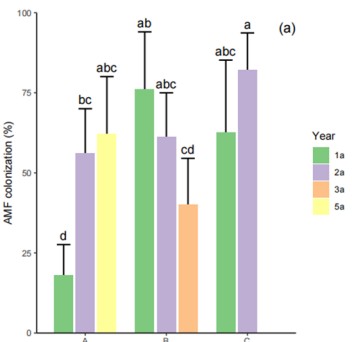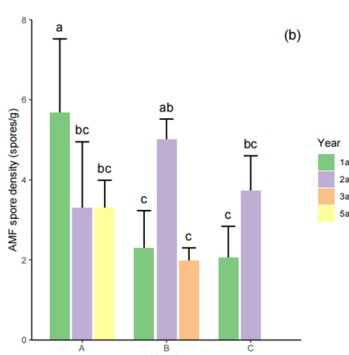

**Figure 1.** AMF colonization (**a**) and spore density (**b**) of *S. grosvenorii*. Differences between the groups are denoted by letters above the bar (analyzed using one-way ANOVA); significance level was set at 5%. A, B, and C represent low (200–300 m), middle (500–600 m), and high (700–800 m) altitudes, respectively. Years represent continuous cropping years (**a**). AMF: arbuscular mycorrhizal fungi.

The average colonization of AMF in the roots of *S. grosvenorii* at low altitudes increased with increasing continuous cropping years in the following order: A1 (18.00%) < A2 (56.00%) < A5 (62.00%). Conversely, spore density at low altitudes decreased with increasing continuous cropping years in the following order: A1 (5.680/g) > A2 (3.304/g) > A5 (3.296/g). The root AMF colonization of *S. grosvenorii* planted for the first time was substantially lower in the low-altitude treatment than in the middle- and high-altitude treatments; however, the AMF spore density in the rhizosphere soil was significantly higher in the low-altitude treatment than in the middle- and high-altitude treatments. In the low-altitude area, the colonization of *S. grosvenorii* planted for the first time was significantly lower than that planted for 2 and 5 years; in contrast, the spore density of AMF in the rhizosphere soil showed an opposite trend, and the spore density of *S. grosvenorii* planted for the first time was significantly higher than that planted in the second and fifth years.

### 4.3. AMF Community Composition in Rhizosphere Soil of S. grosvenorii

A total of 2,968,718 raw reads were obtained from the rhizosphere soil samples of *S. grosvenorii*; 2,961,295 clean total reads were obtained after filtering and optimizing low-quality reads; and 2,530,204 clean total tags were obtained with splicing quality control. Comparing the AMF reads with the database, 2948 OTUs and 2,460,139 sequences were obtained.

AMF OTUs primarily belonged to *Paraglomeraceae* (221 OTUs, 7.1%), *Glomeraceae* (100 OTUs, 3.3%), *Claroideoglomeraceae* (16 OTUs, 0.5%), *Acaulosporaceae* (8 OTUs, 0.2%),

*Ambisporaceae* (7 OTUs, 0.2%), *Diversisporaceae* (4 OTUs, 0.1%), *Gigasporaceae* (4 OTUs, 0.1%), and *Archaeosporaceae* (2 OTUs, <0.1%). AMF species in the rhizosphere soil of *S. grosvenorii* included *Glomus*, *Claroideoglomus*, *Acaulospora*, *Paraglomus*, *Ambispora*, *Archaeospora*, *Diversispora*, *Gigaspora*, and *Scutellospora*. The genus and species information are presented in Table 2. The relative abundance of the AMF community in the rhizosphere soil of *S. grosvenorii* is shown in Figure 2. *Glomus* (average, 13.278%), *Paraglomus* (7.175%), *Claroideoglomus* (0.702%), *Ambispora* (0.278%), *Acaulospora* (0.124%), *Gigaspora* (0.010%), and *Diversispora* (0.003%) had the highest relative abundances. *Paraglomus* was the dominant AMF genus in the rhizosphere soil of *S. grosvenorii* planted for the first time at low altitudes; *Glomus* and *Paraglomus* were the dominant genera at middle and high altitudes. *Glomus* and *Paraglomus* were the dominant genera in the rhizosphere soil of *S. grosvenorii* planted for 2 and 5 years in low-altitude areas.

**Table 2.** Statistics of AMF composition in rhizosphere soil of *S. grosvenorii*.

| Order | Family | Genus | Species |
|---|---|---|---|
| *Archaeosporales* | *Ambisporaceae* | *Ambispora* | *A. leptoticha* |
| | *Archaeosporaceae* | *Archaeospora* | *Ar. Other1* |
| | *Acaulosporaceae* | *Acaulospora* | *Ac. Acau10, Acaulospora* sp. |
| *Diversisporales* | *Diversisporaceae* | *Diversispora* | *Diversispora* sp. |
| | *Gigasporaceae* | *Gigaspora* | *Gi. decipiens* |
| | | *Scutellospora* | *S. heterogama* |
| *Glomerales* | *Claroideoglomeraceae* | *Claroideoglomus* | *C. Douhan9, C.GlBb12, C.lamellosum, C.ORVIN_GLO4, C. Torrecillas12b_Glo_G5* |
| | *Glomeraceae* | *Glomus* | *G.Alguacil09b_Glo_G3, G.caledonium, G. clarum, G.Glo3b, G. Glo49, G.MO_G17, G.MO_G18, G.MO_G40, G.ORVIN_GLO1E, G.ORVIN_GLO3B, G.ORVIN_GLO3D, G.ORVIN_GLO3E, G. Torrecillas12b_Glo_G13, G. viscosum, G. Wirsel_OTU16, G. Yamato09_E* |
| *Paraglomerales* | *Paraglomeraceae* | *Paraglomus* | *P. Alguacil12a_Para_1, P. Alguacil12b_ACA1* |

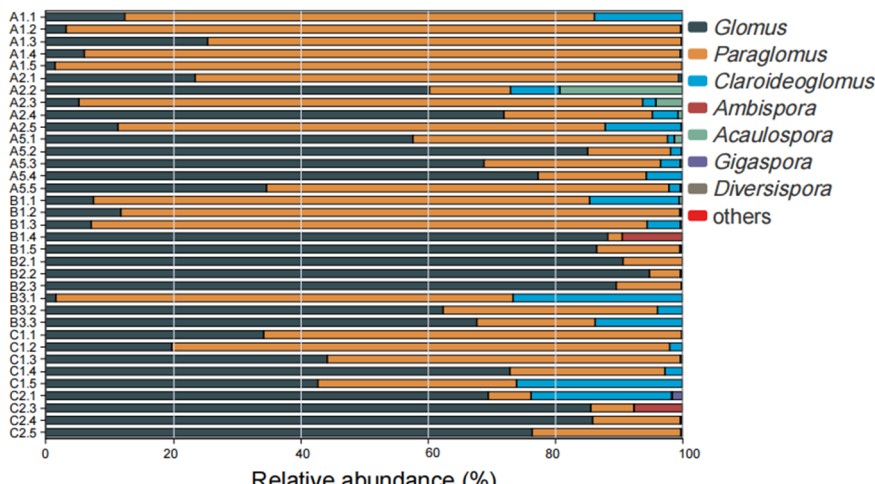

**Figure 2.** Relative abundance of AMF community in the rhizosphere soil of *S. grosvenorii*. AMF: arbuscular mycorrhizal fungi.

*4.4. AMF Diversity in Rhizosphere Soil of S. grosvenorii*

The AMF community richness index ranged from 355 to 762, and the Chao1 index ranged from 356.8 to 763.2. The Shannon index of AMF community diversity ranged

from 0.894 to 2.09; the Simpson index ranged from 0.0324 to 0.328. An NMDS analysis was performed on the samples using the Bray–Curtis distance algorithm (Figure 3). The species composition of each sample group was different, and the differences in the AMF community structure in the rhizosphere soil of *S. grosvenorii* were affected by altitude. The Stress parameter was 0.1810, indicating that the NMDS analysis is reliable.

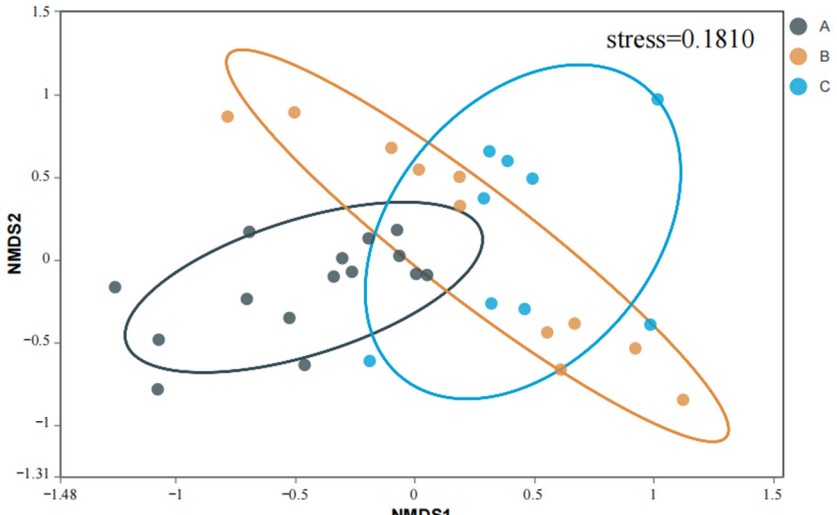

**Figure 3.** NMDS analysis of AMF in rhizosphere soil of *S. grosvenorii*. A, B, and C represent low (200–300 m), middle (500–600 m), and high (700–800 m) altitudes, respectively. NMDS: non-metric multidimensional scaling, AMF: arbuscular mycorrhizal fungi.

Based on the OTU level and Bray–Curtis distance algorithm, we performed AMOVA and ANOSIM. Significant differences between the low-, middle-, and high-altitude groups were noted, and between-group differences were significantly higher than within-group differences. Therefore, the grouping was significant and confirmed the NMDS results.

The Venn analysis (Figure 4) showed 730 shared OTUs in *S. grosvenorii* at low, middle, and high altitudes, with 637, 459, and 262 unique OTUs, respectively. The LEfSe analysis was performed on the genera and species, which showed significant differences in the relative abundance of AMF in the rhizosphere soil of *S. grosvenorii* planted for the first time at different altitudes and continuous cropping years at low altitudes (Figure 5); biomarkers with statistical differences were discussed.

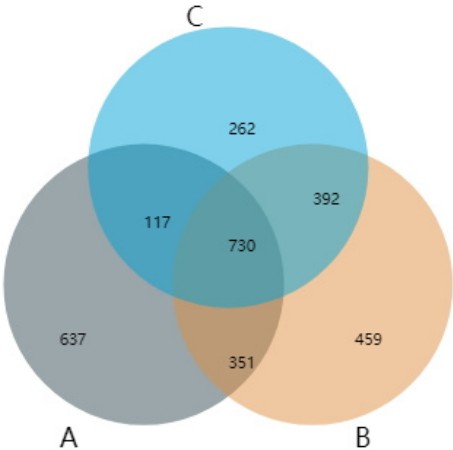

**Figure 4.** Venn diagram of AMF in *S. grosvenorii* rhizosphere soil. Different colors represent different groups. The overlapping region represents the number of OTUs shared by the group, and the non-overlapping region represents the number of OTUs specific to the group. A, B, and C represent low (200–300 m), middle (500–600 m), and high (700–800 m) altitudes, respectively. AMF: arbuscular mycorrhizal fungi.

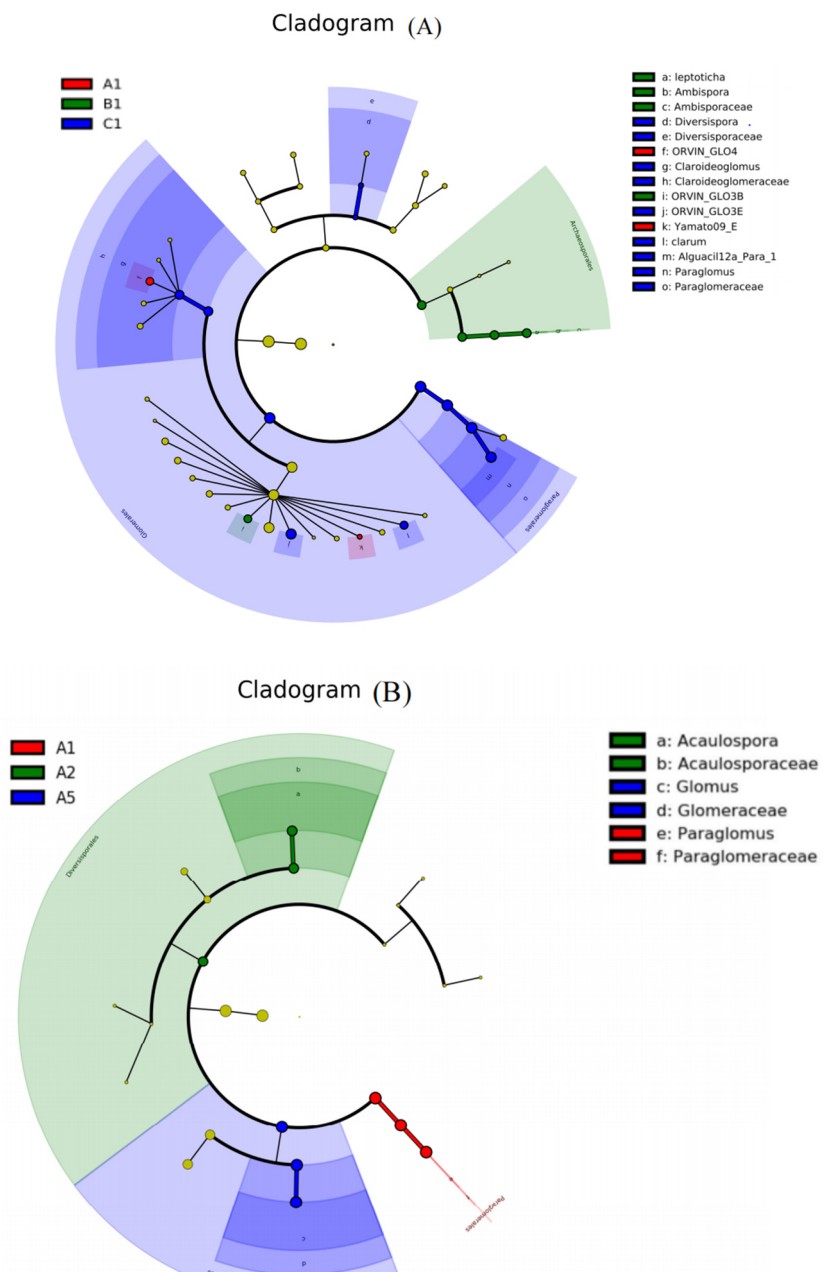

**Figure 5.** LEfSe analysis of AMF in rhizosphere soil of *S. grosvenorii* first planted at different altitudes (**A**) and different continuous cropping years at low altitudes (**B**). A, B, and C represent low (200–300 m), medium (500–600 m), and high (700–800 m) altitudes, respectively; the numbers represent continuous cropping years. A1, A2, and A5 represent low-altitude (200–300 m) continuous cropping for 1, 2, and 5 years, respectively. B1 represents continuous cropping for 1 year at a medium altitude (500–600 m). C1 represents 1 year continuous cropping at a high altitude (700–800 m). LEfSe: linear discriminant analysis effect size; AMF: arbuscular mycorrhizal fungi.

Differences in the AMF community composition were observed in the rhizosphere soil of *S. grosvenorii* planted for the first time at different altitudes. Species with significant differences in the relative abundance of AMF in the rhizosphere soil of *S. grosvenorii* planted for the first time at different altitudes were screened. These species included *Glomus Yamato09_E* and *Claroideoglomus ORVIN_GLO4* at low altitudes; *Glomus ORVIN_GLO3B* and Ambispora leptoticha at middle altitudes; and *Glomus clarum*, *Glomus ORVIN_GLO3E*, and *Paraglomus Alguacil12a_Para _1* at high altitudes.

Differences in AMF community composition were observed in the rhizosphere soil of *S. grosvenorii* with different continuous cropping years at low altitudes. At the genus level, species with significant differences in the relative abundance of AMF in the rhizosphere soil of *S. grosvenorii* planted at low altitudes for 1, 2, and 5 years were screened. The species with significant differences in the first year and at 2 and 5 years of planting at low altitudes were *Paraglomus*, *Acaulospora*, and *Glomus*, respectively.

### 4.5. Effects of Environmental Factors on AMF Community Composition of S. grosvenorii

DCA showed that the length of DCA1 (axis lengths) was >4, indicating that the CCA was more suitable than DCA for reflecting the relationship between microbial flora, samples, and environmental factors. CCA1 and CCA2 explained 21.7% and 18.1% of the total variation, respectively (Figure 6). Additionally, altitude was positively correlated with the organic matter content.

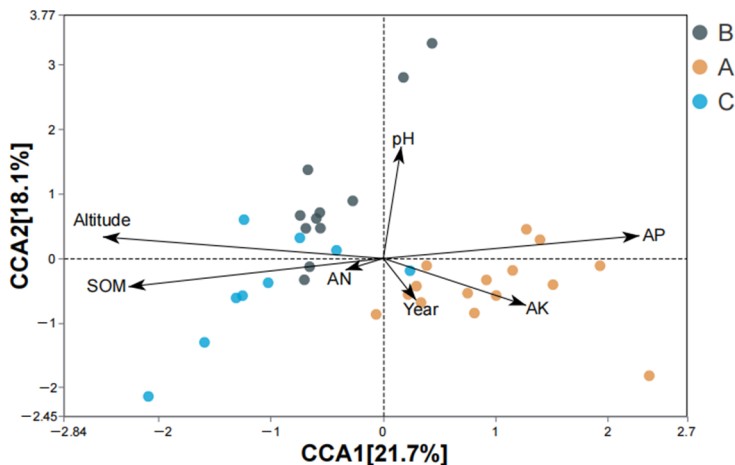

**Figure 6.** CCA of environmental factors and AMF community. A, B, and C represent low (200–300 m), medium (500–600 m), and high (700–800 m) altitudes, respectively. CCA: canonical correspondence analysis; AMF: arbuscular mycorrhizal fungi. SOM: soil organic matter. AN: available nitrogen. AP: available phosphorus. AK: available potassium.

According to the results of the Mantel analysis (Table 3), altitude, organic matter, and AP had significant effects on AMF community composition; however, other factors, such as continuous cropping years, AN, and AK had no significant effect. A heat map (Figure 7) was drawn based on the Spearman correlation coefficient between environmental factors and the abundance of microbial flora at the genus level. Altitude was positively correlated with the relative abundances of *Glomus* and *Ambispora*, whereas the AP content was negatively correlated with that of *Glomus*, *Ambispora*, *Paraglomus*, *Claroideoglomus*, *Diversispora*, and *Scutellospora*.

**Table 3.** Mantel analysis of environmental factors and AMF community.

| Factor | *r* | *p*-Value |
|---|---|---|
| Year | −0.12 | 0.959 |
| Altitude | 0.29 ** | 0.001 |
| pH | 0.07 | 0.16 |
| SOM | 0.19 * | 0.01 |
| AN | −0.01 | 0.572 |
| AP | 0.29 ** | 0.001 |
| AK | 0.09 | 0.142 |

Note: * $p < 0.05$. ** $p < 0.01$. AMF: arbuscular mycorrhizal fungi. SOM: soil organic matter. AN: available nitrogen. AP: available phosphorus. AK: available potassium.

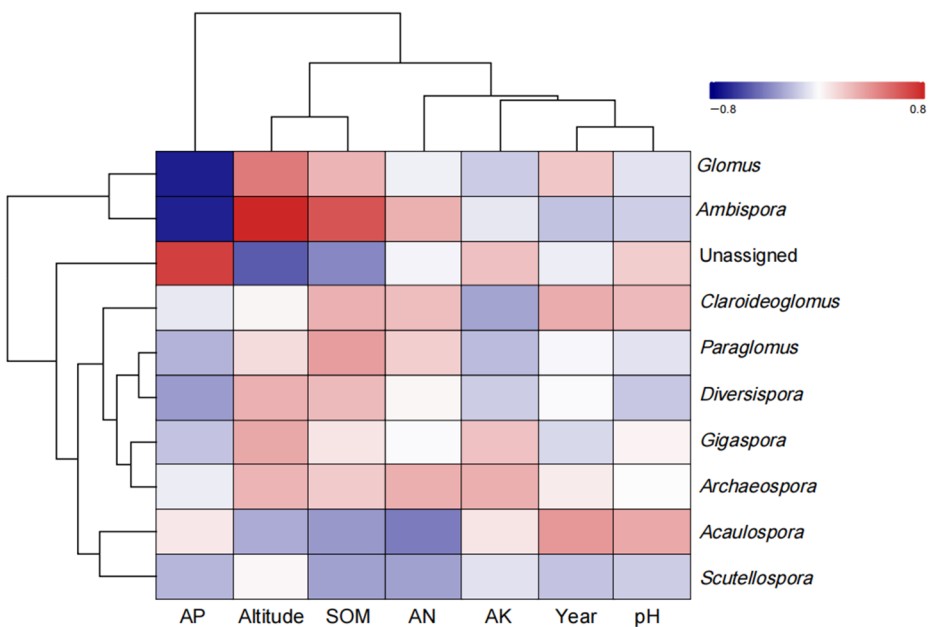

**Figure 7.** Correlation heat map of AMF and environmental factors at the genus level. Red represents a positive correlation and blue represents a negative correlation. The deeper the color, the higher the correlation. AMF: arbuscular mycorrhizal fungi. SOM: soil organic matter. AN: available nitrogen. AP: available phosphorus. AK: available potassium.

*4.6. Relationships between Environmental Factors and AMF Parameters of S. grosvenorii*

The correlations between environmental factors and AMF are summarized in Table 4. Altitude was positively correlated with AMF colonization ($p < 0.01$) and negatively correlated with spore density ($p < 0.05$). AP was significantly negatively correlated with AMF colonization ($p < 0.01$). AN and SOM were positively correlated with AMF colonization, albeit nonsignificantly. Continuous cropping years were positively correlated with the AMF colonization of *S. grosvenorii* roots, albeit nonsignificantly.

**Table 4.** Pearson correlation coefficients of environmental factors with AMF parameters.

|  | Year | Altitude | pH | SOM | AN | AP | AK |
|---|---|---|---|---|---|---|---|
| AMF colonization | 0.084 | 0.454 ** | 0.317 | 0.289 | 0.004 | −0.455 ** | −0.013 |
| Spore density | −0.065 | −0.358 * | −0.314 | −0.371 * | 0.137 | 0.353 * | 0.231 |
| Shannon | 0.145 | −0.051 | 0.381 * | 0.116 | 0.141 | 0.222 | −0.145 |
| Chao1 | 0.206 | −0.189 | 0.346 * | 0.003 | −0.022 | 0.125 | −0.266 |
| Simpson | −0.103 | 0.007 | −0.292 | −0.11 | −0.212 | −0.192 | −0.011 |

* Correlation significant at 0.05. ** Correlation significant at 0.01. AMF: arbuscular mycorrhizal fungi. SOM: soil organic matter. AN: available nitrogen. AP: available phosphorus. AK: available potassium.

## 5. Discussion

*5.1. Medicinal Plant Mycorrhizal Fungi State*

This study investigated and analyzed the AMF infection status of *S. grosvenorii* roots in hilly areas of Guangxi, China for the first time. *S. grosvenorii* can form a good mycorrhizal symbiotic relationship with AMF. Differences in AMF colonization were present in roots of different medicinal plants. The average colonization of AMF in the roots of *S. grosvenorii* was 57.33%, much higher than that of *Houttuynia cordata* Thunb. (4.5%), *Hibiscus mutabilis* L. (13.5%), and *Rosa laevigata* Michx. (16.4%) [23]. However, compared with that of *Michelia figo*. (71.5%), *Rhus chinensis*. (88.5%), *Mirabilis jalapa* Linn. (84%), and *Camptotheca acuminata* Decne. (100%) [23], the AMF colonization of *S. grosvenorii* roots was low. Differences in the abundance of AMFs were observed in the rhizosphere of different medicinal plants. In

a study on the AMF community composition in the rhizosphere soil of the above seven medicinal plants, 26 species of AMFs were present in the rhizosphere of *Rhus chinensis*; however, only nine species of AMFs were observed in the rhizosphere of *Houttuynia cordata Thunb.* [23]. In the present study, the number of AMF species in the rhizosphere of *S. grosvenorii* was 28. Furthermore, *Glomus* was distributed in the rhizosphere soil of the above seven medicinal plants, with the largest number of individual species [23]. This result is consistent with the results of the largest number of *Glomus* species in the rhizosphere of *S. grosvenorii*. Differences in AMF endemic species were present in the rhizosphere of different medicinal plants. *Gigaspora* only appeared in the soil of *Houttuynia cordata Thunb.* and *Scutellospora* only appeared in the soil of *Rosa laevigata Michx.* [23]. However, both *Gigaspora* and *Scutellospora* exist in the rhizosphere soil of *S. grosvenorii*. Based on the above research results, we speculate that the reasons for the differences in AMF colonization, rhizosphere AMF species number, and endemic species of different medicinal plants may be related to various factors, including host plant preference, plant root structure, secretory substances, climate environment, and soil factors [23]. This finding also shows the host diversity of AMFs. In the future, the research scope should be expanded to explore the mycorrhizal dynamics of medicinal plants and the differential response mechanism with external factors.

### 5.2. AMF Community Composition in Rhizosphere Soil of S. grosvenorii

In this study, high-throughput sequencing technology was used to explore the AMF germplasm resources in the rhizosphere soil of *S. grosvenorii*. The results showed that AMF OTUs primarily belonged to *Paraglomeraceae*, *Glomeraceae*, *Claroideoglomeraceae*, *Acaulosporaceae*, *Ambisporaceae*, and *Diversisporaceae*, among others. *Glomus* in the rhizosphere soil of *S. grosvenorii* was related to altitude, continuous cropping years, and soil factors; the number of *Glomus* species, which was 16, as well as its average relative abundance detected in the rhizosphere soil of *S. grosvenorii* were the highest among all species. These results may be related to the symbiotic and reproductive strategies of *Glomus*, which has attracted much attention owing to its broad-spectrum ecological adaptability. *Glomus* can produce more propagules (hyphae, etc.) colonized in plant roots and propagules (spores, etc.) distributed in plant rhizosphere soil [24]. *Glomus* can form a large network of hyphae and adapt to a wide range of soil pH [25], and it has a rapid regeneration ability, high mycelium turnover rate [26], strong anti-interference ability, and strong tolerance, and can rapidly reconstruct mycelium even after interference [27]. Furthermore, *Glomus* helps the host to better cope with complex and changeable environmental factors and better protect the host plants. Thus, the broad-spectrum ecological adaptability of *Glomus* results in the widespread distribution of AMFs in arid areas [28], tropical rainforests [29], deserts [30], and other habitats with different climatic conditions.

### 5.3. Effects of Altitude on AMF Community Composition in S. grosvenorii Roots

The average colonization of AMF in the roots of *S. grosvenorii* planted for the first time was significantly lower in the low-altitude area than in the middle- and high-altitude areas. AMF spore density in the rhizosphere soil was significantly higher in the low-altitude treatment than in the middle- and high-altitude treatments. These results may be attributed to the changes in the dominant AMF genera in the rhizosphere soil of *S. grosvenorii*. *Paraglomus* was the dominant genus of AMF in the rhizosphere soil of *S. grosvenorii* planted for the first time at low altitudes. *Glomus* and *Paraglomus* were the dominant genera at middle and high altitudes; the average relative abundance of *Glomus* increased with an increasing altitude. The average relative abundances of *Glomus* in the rhizosphere soil of *S. grosvenorii* planted for the first time in the middle- and high-altitude areas were higher than those in low-altitude areas; the treatment of continuous cropping for 2 years had similar results.

Four possible reasons can explain these findings. The first may be related to the low-temperature environments at high altitudes. With a 3 °C temperature increase, AMF

colonization decreases significantly [31,32], and the spore density of AMF is affected by temperature. High temperatures and light intensities can increase AMF spore production [27,33,34], indirectly indicating that the low-temperature environment caused by a high altitude significantly contributes to AMF colonization but is not conducive to AMF sporulation. The second reason may be related to the change in soil nutrients along the altitude gradient. The AMF colonization of *S. grosvenorii* roots was positively correlated with the SOM content and negatively correlated with the AP content. A factor analysis of rhizosphere soil of *S. grosvenorii* planted for the first time and continuously cropped for 2 years showed that with an increase in altitude, the content of SOM increased, whereas the AP content decreased; the changing trend in soil nutrient conditions contributed to AMF infection. The third reason may be related to the survival strategy of the broad-spectrum ecologically adaptive symbiotic strain *Glomus* [35]. *Glomus* is widely distributed in various habitats with different climatic conditions, and it has a strong tolerance and resistance to complex environmental factors and can adapt to soil fertility conditions. The pH value of the 200–800 m elevation *S. grosvenorii* planting area and the relative abundances of *Glomus* in the rhizosphere soil of *S. grosvenorii* planted at altitudes of 500–600 m and 700–800 m for the first time were higher than those at an altitude of 200–300 m. Lastly, the fourth reason may be related to an increased AMF colonization at elevated altitudes. A significant positive correlation was observed between altitude and AMF colonization, which was similar to that for soybean [36] and *Artemisia ordosica* [37]. The relative abundances of AMF species *Glomus Yamato09_E*, *Glomus ORVIN_GLO3B*, *Glomus clarum*, *Glomus ORVIN_GLO3E*, and *Paraglomus Alguacil12a_Para_1* in the rhizosphere soil of *S. grosvenorii* planted for the first time were different at different altitudes, which changed the AMF community structure.

An increase in altitude led to a decrease in regional environmental temperature, an increase in the SOM content, and a decrease in the AP content. These factors improved the AMF infection ability, affected the AMF species abundance, and dominant genera and species compositions in the rhizosphere soil of *S. grosvenorii*. This phenomenon in turn resulted in differences in root colonization and soil spore density. These differences were more distinctly obvious at altitudes below 300 and 700 m.

### 5.4. Effects of Continuous Cropping Years on AMF Community in S. grosvenorii

In the low-altitude area, the colonization of *S. grosvenorii* planted for the first time was significantly lower than that of continuous cropping for 2 and 5 years, whereas the spore density of AMF in the rhizosphere soil showed a contrary trend. The AMF spore density in *S. grosvenorii* planted for the first time was significantly higher than that after 2 and 5 years. This change was also related to the change in the AMF species composition in the soil. *Paraglomus* was the dominant genus in the rhizosphere of *S. grosvenorii* planted for the first time in the low-altitude area. *Glomus* and *Paraglomus* were the dominant genera in the rhizosphere soil of *S. grosvenorii* planted for 2 and 5 years. With an increase in continuous cropping years, the average relative abundance of *Glomus* increased.

Four possible reasons could explain these findings. First, continuous cropping affects the nutrient levels of the plant rhizosphere soil [36]. Owing to fertilizer use, with an increase in continuous cropping years in low-altitude areas, the nutrient level of the rhizosphere soil of *S. grosvenorii* increased, SOM accumulation increased, and AP content decreased. SOM and AP contents were positively correlated and significantly negatively ($p < 0.01$) correlated with the AMF colonization of *S. grosvenorii* roots, respectively. The increase in continuous cropping years led to an increase in the SOM content and a decrease in the AP content, which contributed to AMF colonization in the roots of *S. grosvenorii*. The second reason may be related to the soil tillage method used in the planting area of *S. grosvenorii*. The planting area of *S. grosvenorii* is subjected to a non-mechanical, non-organic, and traditional agricultural soil tillage method, which is beneficial for increasing microbial community diversity in plant rhizosphere soil [38]. However, soil tillage may reduce the spore production of some AMF species (such as non-*Glomus*) [39], which explains the positive correlation between continuous cropping years and the AMF diversity of

*S. grosvenorii* and negative correlation between continuous cropping years and spore density. Third, the continuous cropping of plants may affect the soil microbial community structure for many years [36]. It transforms the soil microbial community structure from "bacterial" to "fungal" [36], and the change in soil microbial community structure may be conducive to AMF colonization [35,40]. Lastly, the fourth reason may be related to the reproductive strategy of *Glomus*, which is widely distributed in diverse ecosystems. *Glomus* can produce a significant number of propagules that colonize plant roots (hyphae) and are distributed in the plant rhizosphere soil (spores) [24]. *Glomus* can form an extensive network of hyphae, adapt to a wide range of soil pH [25], and result in a positive correlation between the continuous cropping years of *S. grosvenorii* and the relative abundance of *Glomus*. With an increase in continuous cropping years, the average relative abundance of *Glomus* in the rhizosphere soil of *S. grosvenorii* planted at an altitude of 200–300 m also increased.

In low-altitude areas (200–300 m), continuous cropping affected the relative abundances of the AMF species *Paraglomus* and *Glomus* in the rhizosphere soil of *S. grosvenorii* and changed the dominant genera and community composition of soil AMF, leading to differences in the colonization and community composition of AMF in different continuous cropping years. Furthermore, with an increase in continuous cropping years, the root AMF colonization of *S. grosvenorii* increased at an altitude of 200–300 m.

### 5.5. Effects of Soil Factors on AMF of S. grosvenorii

Soil phosphorus was the primary factor affecting the AMF colonization of *S. grosvenorii* roots. In this study, AP was significantly negatively correlated with AMF colonization and was also negatively correlated with the relative abundances of *Glomus* and *Paraglomus*; however, AK was not significantly correlated with AMF colonization. Soil phosphorus primarily originates from rock weathering and fertilization [41]. Studies suggest that plant roots absorb phosphorus via two main pathways. One pathway is chosen when the soil is phosphorus-limited; in this case, plants strengthen their cooperation with AMFs and become highly dependent on it; plants thereby use the AMF hyphae of plant roots to absorb phosphorus [42]. Under a soil phosphorus limitation, mycorrhiza secretes organic acids and $H^+$ to promote mineral weathering, which in turn promotes plant phosphorus uptake [42]. Simultaneously, plants tend to allocate carbon to AMFs, and the carbon content provided by plants to AMFs is proportional to the phosphorus content provided by AMFs [43]. Therefore, plants and AMFs are mutually beneficial. Another pathway is chosen when the soil is phosphorus-rich, and plants use their roots to directly absorb phosphorus [42]. Consistent with the results of previous studies [32,44,45], we found that excessive soil nutrients, such as AP, may result in unproductivity or exhibit adverse effects on AMF. In this study, soil phosphorus was negatively correlated with AMF abundance and colonization as fertilization increases the phosphorus content of plants and decreases root cell membrane permeability, thereby reducing the chance of a secondary invasion of AMF in roots [44]. Moreover, the phosphorus obtained by fungi cannot be transmitted to the host early; therefore, the rate of the turnover of phosphorus in the hyphae slows down, resulting in the accumulation of fungal nutrients and imbalance of metabolic regulation. This phenomenon affects the growth of fungi, limits the nutrient exchange between plant roots and AMF, and reduces mycorrhizal colonization and the mycorrhizal effect [44,46]. Therefore, excessive phosphorus content affects the resource exchange between plants and fungi, causing a change in the AMF community in the rhizosphere soil, negatively affecting AMF colonization. Therefore, in the process of *S. grosvenorii* plant cultivation, by regulating the soil phosphorus level, ensuring the promoting effect of plant rhizosphere beneficial symbiotic bacteria on plant nutrient absorption, and using species association can result in less resource usage and the promotion of green development.

Acidic soils support AMF infection and increase the AMF diversity index of *S. grosvenorii* roots. Soil pH is an important factor in determining the AMF niche and regulating community composition [47]. Slightly acidic soils have been suggested as suitable for the survival of AMF as extreme pH affects fungal colonization [48]. In contrast, alkaline soils may cause

some species to disappear [49]. In the study of *S. grosvenorii*, the average pH value of rhizosphere soil was 5.00, and no extreme alkaline pH was present. The weak acidic soil pH of *S. grosvenorii* increased the abundance of some non-dominant genera, *Claroideoglomus* and *Acaulospora*, in the rhizosphere soil, which supported the claims that acidic soil is beneficial to the survival of *Acaulosporaceae* [26], and that acidic soil *Acaulospora* sporulation was more abundant [23]. Therefore, weakly acidic soil may be beneficial to the germination and infection of AMFs in the rhizosphere of *S. grosvenorii* and increase the colonization of AMF. This finding is similar to the results of a study that showed that pH was positively correlated with the mycorrhizal colonization of *Paris polyphylla* var. yunnanensis at the fruit ripening stage [50], and pH was positively correlated with the AMF diversity of 28 plants [47]. AMFs form hyphae inside and outside the root, and pH affects the niche space of AMFs [49]. However, some studies have suggested that AMF is suitable for alkaline soil [48]. Therefore, soil pH is an important driving factor affecting the abundance and diversity of microbial communities [34]. The AMF community was constructed based on pH niche differentiation [47]. However, the optimal soil pH range of AMF is varied, and no unified conclusion exists on the positive and negative driving effects of soil pH on fungal infection [31]. The mechanism of soil pH affecting the dynamics of the AMF community requires further clarification, which also reflects the sensitivity of the microbial community to environmental factors.

Organic matter and AN contents in rhizosphere soil could facilitate the AMF infection of *S. grosvenorii* roots. This study showed that both AN and SOM were positively correlated with AMF colonization, and that AN and SOM had a positive effect on the relative abundances of *Ambispora*, *Claroideoglomus*, *Paraglomu*, and *Archaeospora*, and they were negatively correlated with the relative abundances of *Acaulospora* and *Scutellospora*. A similar effect may exist between SOM and AN soil nutrients. Researchers generally believe that the AMF colonization of plant roots is related to the turnover of C and N [34,51]. When N content is sufficient, plant growth is promoted and plant carbon sequestration potential is enhanced [26], which in turn contributes to organic carbon accumulation, increased carbon production for AMFs, and AMF colonization [34]. To a certain extent, SOM can promote the growth of AMFs as a substrate for AMF hyphae preservation [52]. Furthermore, the number of AMFs (e.g., frequency of AMF species [39]) increases with an increasing SOM content [53]; AN also has a similar effect. Soil nutrient changes affected the composition of the AMF community, which reduced or removed fungal species, such as *Acaulospora* and *Scutellospora*, and increased the abundance and dominance of fungal species, such as *Glomus* and *Claroideoglomus*. The study also found that SOM was positively correlated with the relative abundance of *Gigaspora* and *Scutellospora* in different genera of the same family (*Gigasporaceae*). The perception ability and sensitivity of different AMFs to the same soil nutrient condition may be different. Nitrogen enrichment promotes plants to transfer more carbon to *Gigasporaceae*, resulting in a higher *Gigasporaceae* carbon sink [26]. The effect of SOM on *Gigasporaceae* may be achieved by regulating the abundance of the two genera (*Gigaspora*, *Scutellospora*) of the family to achieve an effective utilization of soil resources and maximize host benefits [26]. SOM affected the composition of the AMF community and abundances of *Glomus* and *Paraglomus* in the rhizosphere soil of *S. grosvenorii,* and it promoted the spore germination and infection of plant roots, resulting in changes in colonization, community richness, and diversity. This finding is similar to the results obtained for mango litchi orchards [14], *Ammopiptanthus mongolicus* [53,54], *Artemisia argyi* [55], and *Vaccinium uliginosum* [56].

## 6. Conclusions

In this study, high-throughput technology was used to study mycorrhizal fungal species resources and the community composition of *S. grosvenorii* in the hilly areas of Guangxi, China. The results showed that AMF species in the rhizosphere soil of *S. grosvenorii* included 28 species of nine genera, including *Glomus*, *Claroideoglomus*, *Acaulospora*, *Paraglomus*, *Ambispora*, and so on. Altitude, AP, organic matter, and pH

were important factors for predicting the AMF infection and community structure and dynamics of *S. grosvenorii*. The results provided a theoretical basis for the future isolation and screening of dominant strains, inoculation of medicinal plant mycorrhizal fungi, response mechanism and experimental research of mycorrhizal fungi and the soil nutrient gradient, prediction of biodiversity models, and strengthening of the ecological function application of mycorrhizal fungi.

This study has some limitations, including a small experimental area, single-test variety, short continuous cropping time, and single experimental method. Therefore, the specific response mechanism of strains and the environment and the relationship between different AMF species and the selection preference of hosts remain unclear. In future research, morphology, metagenomics, molecular genetics, and other technical methods should be used in the study of AMF diversity and community dynamics from morphological, molecular, and other aspects; the investigation scope of mycorrhizal fungi germplasm resources should be expanded; and the relationship between soil nutrients, rock-parent materials, climatic conditions, organic fertilizer types and dosage, host species, the active ingredient content of medicinal plants, inoculation time, and other multiple factors and flora should be considered to quantitatively analyze rhizosphere mycorrhizal dynamics and the species association of medicinal plants.

**Author Contributions:** Conceptualization, Z.Z. and L.Y.; methodology, Z.Z. and L.Y.; software, L.Y.; validation, L.Y., Z.Z., L.Z. and K.H.; formal analysis, L.Y. and Z.Z.; investigation, L.Y., Z.Z., L.Z. and K.H.; resources, L.Y., Z.Z., L.Z. and K.H.; data curation, Z.Z. and L.Y.; writing—original draft preparation, L.Y.; writing—review and editing, Z.Z. and L.Y.; visualization, L.Y.; supervision, Z.Z.; project administration, Z.Z.; funding acquisition, Z.Z. All authors have read and agreed to the published version of the manuscript.

**Funding:** This work was funded by the National Natural Science Foundation of China (grant numbers 31960272 and 41603079) and the Basic Research Fund of Guangxi Academy of Sciences (grant number CQZ-E-1909). Funder: Zhang, Z.

**Institutional Review Board Statement:** Not applicable.

**Data Availability Statement:** Not applicable.

**Conflicts of Interest:** The authors declare no conflict of interest.

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
