# Peer review of "Effects of Altitude and Continuous Cropping on Arbuscular Mycorrhizal Fungi Community in Siraitia grosvenorii Rhizosphere"

_agriculture, doi:10.3390/agriculture13081548_

Round 1

Reviewer 1 Report

Comments

1. The dominance of Glomus and Paraglomus in the rhizosphere soil is interesting. Can you provide some insights into the ecological significance of these dominant genera and their potential roles in the symbiotic relationship with S. grosvenorii?

2. The study mentions that the slightly acidic rhizosphere soil contributed to AMF colonization and improved richness and diversity. Can you provide more information on how the soil pH was determined and its potential impact on AMF colonization and community dynamics?

3. The study found a negative correlation between available phosphorus (AP) content and AMF colonization. Could you provide further discussion on the possible mechanisms underlying this relationship and its implications for S. grosvenorii cultivation?

4. It would be beneficial to discuss the practical implications of the findings in terms of applying AMF in plant cultivation. How can the knowledge gained from this study be translated into agricultural practices to enhance the growth and productivity of S. grosvenorii?

5. Were there any limitations or challenges encountered during the study that might have influenced the results or interpretation of the findings? Please address any potential limitations and suggest areas for future research.

6. In the discussion, it would be valuable to compare the findings of this study with previous research on AMF communities in the rhizosphere of other medicinal plants or crops. Are there any similarities or differences in AMF colonization and community structure that can be observed?

7. The study focuses on altitudes and continuous cropping years. Did you consider any other factors that could potentially influence AMF community dynamics, such as soil moisture, temperature, or plant nutrient requirements? If so, please provide some insights into these factors and their effects on the results.

8. Please ensure that the implications and significance of the study findings are clearly stated in the conclusion. How can this research contribute to the field of plant-microbe interactions and sustainable agriculture?

9. Is the study's investigation of AMF infection in S. grosvenorii roots in Guangxi, China, a significant contribution to the field? Does it provide new insights or information? Provide information in discussion.

10. How does the observed AMF colonization in S. grosvenorii roots compare to other medicinal plants studied in terms of its percentage? What factors may contribute to the variations in mycorrhizal colonization among different medicinal plants? Provide information in discussion.

11. Can you comment on the relationship between altitude and AMF colonization in S. grosvenorii roots? How does the study's findings support the hypothesis that AMF colonization increases with increasing altitude? Provide information in discussion.

12. What are the possible reasons for the differences in AMF colonization and community composition in S. grosvenorii roots at different altitudes? How do temperature, soil nutrients, pH, and Glomus adaptability contribute to these differences? Provide information in discussion.

13. How does continuous cropping affect AMF colonization in S. grosvenorii roots? What are

the reasons behind the changes in AMF species composition and the increase in Glomus relative

abundance with continuous cropping years? Provide information in discussion.

14. Can you elaborate on the role of soil factors, particularly soil phosphorus, in affecting AMF

colonization of S. grosvenorii roots? What mechanisms may explain the negative correlation

between AP content and AMF colonization?

15. What is the significance of the slightly acidic soil pH for AMF infection and colonization in

S. grosvenorii roots? How do these findings align with previous studies on soil pH and AMF

diversity?

16. How do the organic matter and nitrogen content in the rhizosphere soil influence AMF

infection in S. grosvenorii roots? What are the implications of the positive correlations between

SOM and AN contents with AMF colonization and community composition?

17. In your opinion, are there any limitations or potential confounding factors in the study design

or methodology that could affect the interpretation of the results?

18. Based on the findings of this study, what are the practical implications or applications for the

cultivation of S. grosvenorii or other medicinal plants in terms of optimizing AMF colonization

and enhancing plant growth?

19. Are there any suggestions for further research or additional experiments that could build

upon the findings of this study?

minor improvement is required.

Author Response

Response to Reviewer 1 Comments

Point 1: The dominance of Glomus and Paraglomus in the rhizosphere soil is interesting. Can you provide some insights into the ecological significance of these dominant genera and their potential roles in the symbiotic relationship with S. grosvenorii?

Response 1: Thank you for your advice. We agree with the reviewer 's point of view, and have added the ecological significance of important AMF strains and the potential role in the symbiotic relationship of Siraitia grosvenorii.

Point 2: The study mentions that the slightly acidic rhizosphere soil contributed to AMF colonization and improved richness and diversity. Can you provide more information on how the soil pH was determined and its potential impact on AMF colonization and community dynamics?

Response 2: We fully agree with your viewpoint. We have supplemented the relevant information on soil pH and AMF community dynamics in the manuscript to further improve it.

Point 3: The study found a negative correlation between available phosphorus (AP) content and AMF colonization. Could you provide further discussion on the possible mechanisms underlying this relationship and its implications for S. grosvenorii cultivation?

Response 3: The colonization and community dynamics of soil phosphorus and AMF have received much attention. We have supplemented the impact and ecological significance of soil phosphorus on AMF community dynamics.

Point 4: It would be beneficial to discuss the practical implications of the findings in terms of applying AMF in plant cultivation. How can the knowledge gained from this study be translated into agricultural practices to enhance the growth and productivity of S. grosvenorii?

Response 4: Thank you for your advice. We further improved the application, significance and future research direction of Siraitia grosvenorii rhizosphere AMF in the manuscript.

Point 5: Were there any limitations or challenges encountered during the study that might have influenced the results or interpretation of the findings? Please address any potential limitations and suggest areas for future research.

Response 5: Thank you for your advice. In the manuscript, we proposed the shortcomings of this study and the future research direction of mycorrhizal fungi of medicinal plants.

Point 6: In the discussion, it would be valuable to compare the findings of this study with previous research on AMF communities in the rhizosphere of other medicinal plants or crops. Are there any similarities or differences in AMF colonization and community structure that can be observed?

Response 6: We noticed the differences and similarities of AMF colonization and community structure in the rhizosphere of different medicinal plants, and supplemented relevant information in the manuscript for comparison.

Point 7: The study focuses on altitudes and continuous cropping years. Did you consider any other factors that could potentially influence AMF community dynamics, such as soil moisture,  temperature, or plant nutrient requirements? If so, please provide some insights into these factors and their effects on the results.

Response 7: The purpose of this study was to explore the correlation between altitude, continuous cropping years, main physicochemical factors of soil and AMF colonization, germplasm resources and community structure in the rhizosphere of Siraitia grosvenorii. The effects of soil moisture, temperature and nutrient requirements of plants on AMF have important research value. They are temporarily beyond the scope of this study, but very worthy of our discussion in subsequent experimental studies.

Point 8: Please ensure that the implications and significance of the study findings are clearly stated in the conclusion. How can this research contribute to the field of plant-microbe interactions and sustainable agriculture?

Response 8: Thank you for your advice. We have supplemented the significance of the findings in the conclusion.

Point 9: Is the study's investigation of AMF infection in S. grosvenorii roots in Guangxi, China, a significant contribution to the field? Does it provide new insights or information? Provide

information in discussion.

Response 9: This study is the first to use high-throughput technology to investigate the AMF germplasm resources in the rhizosphere of Siraitia grosvenorii in the hilly area of Guangxi, China. We have provided the novelty and significance of the research in the corresponding position in the manuscript.

Point 10: How does the observed AMF colonization in S. grosvenorii roots compare to other medicinal plants studied in terms of its percentage? What factors may contribute to the variations in mycorrhizal colonization among different medicinal plants? Provide information in discussion.

Response 10: We have provided information on AMF colonization and community composition of different medicinal plants in the discussion.

Point 11: Can you comment on the relationship between altitude and AMF colonization in S. grosvenorii roots? How does the study's findings support the hypothesis that AMF colonization

increases with increasing altitude? Provide information in discussion.

Response 11: We combined four aspects: temperature changes in high-altitude areas, differences in soil nutrients, Glomus survival strategy, and AMF infection ability, and explained in detail the possible reasons for the increase in AMF colonization with increasing altitude during the discussion.

Point 12: What are the possible reasons for the differences in AMF colonization and community composition in S. grosvenorii roots at different altitudes? How do temperature, soil nutrients, pH, and Glomus adaptability contribute to these differences? Provide information in discussion.

Response 12: Thank you for your suggestion. We have elaborated in detail on various aspects such as soil nutrients, pH, Glomus characteristics and survival significance during the discussion.

Point 13: How does continuous cropping affect AMF colonization in S. grosvenorii roots? What are the reasons behind the changes in AMF species composition and the increase in Glomus relative abundance with continuous cropping years? Provide information in discussion.

Response 13: Thank you for your suggestion. We have discussed various aspects such as the impact of plant continuous cropping on soil nutrients, soil microbial communities, soil tillage methods, and Glomus' reproductive strategies.

Point 14: Can you elaborate on the role of soil factors, particularly soil phosphorus, in affecting AMF colonization of S. grosvenorii roots? What mechanisms may explain the negative correlation between AP content and AMF colonization?

Response 14: Thank you for your suggestion. We have elaborated on the relationship between soil phosphorus and AMF from multiple aspects, including soil phosphorus sources, soil phosphorus enrichment and the way plants absorb phosphorus under phosphorus limitation, plant demand for AMF under soil phosphorus limitation, and resource exchange between plants and AMF.

Point 15: What is the significance of the slightly acidic soil pH for AMF infection and colonization in S. grosvenorii roots? How do these findings align with previous studies on soil pH and AMF diversity?

Response 15: We explained the effect and significance of pH on the survival and reproduction of AMF species, and re-described the results of the literature.

Point 16: How do the organic matter and nitrogen content in the rhizosphere soil influence AMF infection in S. grosvenorii roots? What are the implications of the positive correlations between SOM and AN contents with AMF colonization and community composition?

Response 16: We analyzed the relationship between SOM and AN, the response of different AMF species to SOM, and the effect of nitrogen on carbon conversion in different genera of the same family, and explained the effects of soil organic matter and nitrogen on AMF colonization and community of Siraitia grosvenorii.

Point 17: In your opinion, are there any limitations or potential confounding factors in the study design or methodology that could affect the interpretation of the results?

Response 17: Thank you for your question. We conducted in-depth reflection on the experimental research and results, and clarified the limitations of the research in the conclusion.

Point 18: Based on the findings of this study, what are the practical implications or applications for the cultivation of S. grosvenorii or other medicinal plants in terms of optimizing AMF colonization and enhancing plant growth?

Response 18: In this study, the effects of altitude, continuous cropping and soil factors on AMF germplasm resources, community structure and colonization in the rhizosphere of Siraitia grosvenorii were investigated. This study provides basic data and reference for the application and practice of mycorrhizal fungi, such as the subsequent isolation and screening of dominant strains, the inoculation of mycorrhizal fungi, and the exploration of the effects of mycorrhizal fungi on the growth and active ingredient content of medicinal plants.

Point 19: Are there any suggestions for further research or additional experiments that could build upon the findings of this study?

Response 19: Thank you for your question. In the discussion and conclusion part, we have supplemented the views on the future research direction and application of mycorrhizal fungi.

Reviewer 2 Report

Line 35-86 The introduction says nothing about previous studies on the same topic (on S. grosvenorii). It is not clear from the introduction what other studies have said about S. grosvenorii mycorrhiza.

Line 127. What mass of roots was taken?

Line 146 AN decode

Line 147 AP decode

Figure 1. two-sided confidence intervals needed

Figure 2. Represent as a heat map

Line 410-416 - refers to results

Line 422. This section also discusses soil factors.

Line 469. This section also discusses soil factors.

408-590 I recommend redoing the discussion and removing the division into sections. The discussion will be more understandable if some general picture is presented in it. Emphasis should be placed on the factors that affect the most significantly. If these factors are related to others, then this should be discussed immediately, without moving it to another paragraph.

Line 596-597 non-specific statement.

Line 606-610. Obvious statement. This research is not required.

Line 591-610. the conclusion is very vague and is a summary of the discussion.

General conclusion

Novelty must be clearly stated in the abstract and conclusion.. The impact of the assessed factors was expected. The discussion section needs to be rewritten to summarize the results. The paper does not mention other studies of S. grosvenorii mycorrhiza. Is this the first study on this object?

Some phrases may be rewritten. Long sentences can be divided into short ones.

Author Response

Response to Reviewer 2 Comments

Point 1: Line 35-86. The introduction says nothing about previous studies on the same topic (on S. grosvenorii). It is not clear from the introduction what other studies have said about S. grosvenorii mycorrhiza.

Response 1: Thank you. This study is the first to use high-throughput technology to investigate the community structure of mycorrhizal fungi germplasm resources of Siraitia grosvenorii in hilly areas of Guangxi, China. Therefore, the introduction does not mention the study of mycorrhizal fungi of Siraitia grosvenorii.

Point 2: Line 127. What mass of roots was taken?

Response 2: Thank you for your questions, we have added information about sampling in the manuscript.

Point 3: Line 146. AN decode.

Response 3: We supplement the full name of the abbreviations in the manuscript.

Point 4: Line 147. AP decode.

Response 4: Thank you for your detailed guidance, we are very sorry, we have carried out the abbreviation correction.

Point 5: Figure 1. two-sided confidence intervals needed.

Response 5: Thank you for your suggestion, we have redrawn and analyzed this part of the picture and content.

Point 6: Figure 2. Represent as a heat map.

Response 6: Thank you for your advice. There is heat map of the correlation between environmental factors and AMF based on the genus level in the manuscript. We believe that the addition of the relative abundance map of AMF can make the expression of the research results more diversified and attract the attention of readers.

Point 7: Line 410-416. refers to results.

Response 7: Thank you for your guidance. We have corrected the expression of this part.

Point 8: Line 422. This section also discusses soil factors.

Response 8: We found that soil nutrients showed a certain trend with the elevation gradient. Therefore, in the relationship between altitude and AMF colonization community, it was mentioned that altitude may affect the AMF community by affecting soil nutrient changes.

Point 9: Line 469. This section also discusses soil factors.

Response 9: We found that continuous cropping of plants has a certain effect on soil nutrients, so we speculate that continuous cropping may affect the AMF community by affecting soil physicochemical properties.

Point 10: Line 408-590. I recommend redoing the discussion and removing the division into sections. The discussion will be more understandable if some general picture is presented in it. Emphasis should be placed on the factors that affect the most significantly. If these factors are related to others, then this should be discussed immediately, without moving it to another paragraph.

Response 10: Thank you for your advice. We agree with your views. We have revised and re-edited the discussion section, and focused on the important factors affecting the AMF community at different levels.

Point 11: Line 596-597. non-specific statement.

Response 11: Thank you for your suggestion, we redacted this section.

Point 12: Line 606-610. Obvious statement. This research is not required.

Response 12: We re-edited the content of this part, and expounded the importance and significance of this research and the future research direction.

Point 13: Line 591-610. the conclusion is very vague and is a summary of the discussion.

Response 13: Thanks for your guidance, we have rewritten the conclusion part, and elaborated the research results, significance, limitations, future research directions and so on.

Point 14: Novelty must be clearly stated in the abstract and conclusion.. The impact of the assessed factors was expected. The discussion section needs to be rewritten to summarize the results. The paper does not mention other studies of S. grosvenorii mycorrhiza. Is this the first study on this object?

Response 14: Thank you for your detailed guidance and advice. We have clarified the novelty and significance of this study in the abstracts and conclusions. This study was the first to use high-throughput technology to investigate AMF colonization, germplasm resources and community dynamics of Siraitia grosvenorii in northern Guangxi, China.

Reviewer 3 Report

The work is a collection of facts  

The sites chosen differ not only in altitude but as one learns from going to the end of paper in management  -  differing fertilization  cropping history etc   are known to influence AM interactions

So you gain information about what AM genus/species are present    this is what the paper achieves  

should be reconstructed as such

also included should be reports on the plants-  spores etc would relate to root structure and function 

there is none of this expected data 

methods outside of the molecular are lacking in detail  

Professional editing  is required 

Author Response

Response to Reviewer 3 Comments

Point 1: Line 67-68. this sentence needs rephrasing and different wording Meilo...  is a nematode     this requires more discussion is this used locally as a disease name? what does disease will be aggravated mean?

Response 1: Thank you for your advice. We reworded this part of the content and removed the relevant information about nematodes. Before this, we assumed that AMF may have an impact on the continuous cropping obstacle and nematodiasis of Siraitia grosvenorii, but this is beyond our research scope.

Point 2: Line 71-72. where are the references.

Response 2: We have supplemented the references in this part.

Point 3: Line 76-77. do not understand writing needs to be edited. 

Response 3: Thanks for your suggestion, we edited the content again.

Point 4: Line 82-84. do not understand hypotheses no links to mechanisms .ie altitude has consequences that lead to changes in AMF colonization.  

Response 4: We agree with your views. We have modified and re-edited the hypothetical content.

Point 5: Line 106. what was previous crops some crops are protective against root knot nematodes

Response 5: The experimental plot was not planted with any other crops before the experiment. We have re-explained the information section of the experimental plot and deleted the relevant content of nematodes.

Point 6: Line 117-118. there needs to be a table with all of these factors very complex the reader has to make such table to understand what all your abbreviations are to be.

Response 6: Thank you for your proposal. We have annotated the tables and sample code in detail to facilitate the reader 's understanding.

Point 7: Line 128. define root and rhizosphere soil. how were they differentiated

Response 7: Thank you for your advice. We have explained the related nouns in the sampling section.

Point 8: Line 132. no counting of nematodes or the galls think there should be.

Response 8: We have deleted information about nematodes, which is beyond our research scope.

Point 9: Line 146. what is AN what is AP etc need more description.

Response 9: We have supplemented the abbreviations involved in the part of soil physicochemical properties.

Point 10: Line 158. how were data normalized

Response 10: During the analysis of AMF colonization and spore density histogram at different altitudes and continuous cropping years, due to the inconsistency of measurement units and numerical levels between the two, and taking into account the data normalization and convenience for readers to read, we separated and independently analyzed AMF colonization and spore density, and tried not to cause readers' misunderstanding, but also convenient for readers to compare the differences of AMF colonization and spore density under different groups.

Point 11: Line 192. ??? what is this

Response 11: UNITE is the ITS species annotation database. In the analysis of microbial diversity, Operational taxonomic units (OTU) is a method of reducing the dimension of complex sequencing data into simple data. The sequences are usually clustered into different OTUs according to the 97 % similarity threshold, and each OTU is usually regarded as a microbial species. In order to obtain the species classification information corresponding to the OTU, a representative sequence is selected for each OTU for species classification annotation, so as to obtain the community composition of each sample. UNITE is a commonly used ITS species annotation database.

Point 12: Line 218. you give values but there are errors that negate the use of such defined numbers

Response 12: We have restated this part of the content.

Point 13: Line 264. what is significant?.

Response 13: The significance of the results of this part has been explained in the subsequent text.

Point 14: Line 336. if you looked at other plants would you get the same type of separation ie  nothing to do with any factors other than the soils differ in microbial content. 

Response 14: Based on Bray cutis distance algorithm, the NMDS analysis of samples is mainly carried out to detect the relationship between the AMF community structure of Siraitia grosvenorii rhizosphere soil and altitude, and provide basis for subsequent screening and analysis of species with significant differences in each group.

Point 15: Line 357. is harvesting time different at the three altitudes different degrees of plant maturity would impact the AM colonization.

Response 15: We agree with the reviewer 's point of view. It is true that plant maturity will affect AMF colonization, but the effect of plant maturity on AMF community and colonization is beyond the scope of this study. According to the actual situation, considering that fruit harvesting will have a great impact on farmers ' life and economy, we take sampling in November. The relationship between plant maturity and AMF community will be considered in future mycorrhizal fungi research.

Point 16: Line 518. this has been known for a long long time. 

Response 16: As the reviewer 's point of view, we also believe that the effect of soil phosphorus on AMF colonization has been studied on different plants. However, this is the first time to explore the AMF germplasm resources in the rhizosphere of Siraitia grosvenorii, and the response of different plants, AMF species and soil elements is unique to a certain extent. Therefore, we believe that it is necessary to explore the relationship between soil phosphorus and AMF community and colonization in Siraitia grosvenorii.

Point 17: Line 524. And if you knew about the fertilization regime then the altitude hypotheses are futile.

Response 17: We thank the reviewer for the suggestion. We have revised the hypothesis.

Point 18: Line 535. acidity has a huge effect on essential and damaging metal bioavailability not discussed.

Response 18: We agree with you that acidity has a great impact on bioavailability. But this is beyond the scope of this study, and we will seriously consider this view in the follow-up study.

Point 19: Line 563. ?  makes no sense. 

Response 19: Thank you for your suggestion. After careful consideration, we have considered that this part of the content deviates from the research topic, and decided to delete this part of the content.

Point 20: Line 568. how is this vague sentence relevant.

Response 20: We decided to delete this part.

Point 21: The work is a collection of facts. The sites chosen differ not only in altitude but as one learns from going to the end of paper in management - differing fertilization cropping history etc   are known to influence AM interactions. So you gain information about what AM genus/species are present this is what the paper achieves should be reconstructed as such also included should be reports on the plants- spores etc would relate to root structure and function there is none of this expected data methods outside of the molecular are lacking in detail. Professional editing is required.

Response 21: Thank you for your advice. We agree with your views and have made comprehensive improvements to the manuscript. This study is mainly to explore the composition and structure of AMF community and root colonization in the rhizosphere of Siraitia grosvenorii under different altitudes and continuous cropping years. The soil fertilization control is consistent and the specific methods have been described in the article. The spore function may be explored in the subsequent screening of dominant strains and mycorrhizal fungi inoculation test, and discussed in combination with the change trend of plant growth, active ingredient content and so on. Thank you again for your careful guidance.

Round 2

Reviewer 2 Report

The authors took into account the comments. Article may be published

Author Response

Thanks for your review.